# Optimization of the 3D Printing Parameters for Tensile Properties of Specimens Produced by Fused Filament Fabrication of 17-4PH Stainless Steel

**DOI:** 10.3390/ma13030774

**Published:** 2020-02-08

**Authors:** Damir Godec, Santiago Cano, Clemens Holzer, Joamin Gonzalez-Gutierrez

**Affiliations:** 1Faculty of Mechanical Engineering and Naval Architecture, University of Zagreb (UNIZAG FSB), 10000 Zagreb, Croatia; 2Polymer Processing, Montanuniversitaet Leoben, 8700 Leoben, Austria; santiago.cano-cano@unileoben.ac.at (S.C.); clemens.holzer@unileoben.ac.at (C.H.); joamin.gonzalez-gutierrez@unileoben.ac.at (J.G.-G.)

**Keywords:** additive manufacturing, fused filament fabrication, stainless steel 17-4PH, green parts, tensile properties, central composite design, optimization

## Abstract

Fused filament fabrication (FFF) combined with debinding and sintering could be an economical process for three-dimensional (3D) printing of metal parts. In this paper, compounding, filament making, and FFF processing of feedstock material with 55% vol. of 17-4PH stainless steel powder in a multicomponent binder system are presented. The experimental part of the paper encompasses central composite design for optimization of the most significant 3D printing parameters (extrusion temperature, flow rate multiplier, and layer thickness) to obtain maximum tensile strength of the 3D-printed specimens. Here, only green specimens were examined in order to be able to determine the optimal parameters for 3D printing. The results show that the factor with the biggest influence on the tensile properties was flow rate multiplier, followed by the layer thickness and finally the extrusion temperature. Maximizing all three parameters led to the highest tensile properties of the green parts.

## 1. Introduction

Additive manufacturing (AM) comprises a group of technologies used to build physical parts by adding material in a layer-by-layer fashion from a computer-aided design (CAD) file, as opposed to subtractive manufacturing methods, such as machining [1]. AM is the term standardized by ISO and ASTM [2]; however, other common names for AM include three-dimensional (3D) printing, layer-based manufacturing, solid freeform fabrication (SFF), and rapid prototyping (RP) [3,4]. Each of these terms highlights a distinct feature of these processes, for example, 3D printing is a common name used by the general public, whereas the term rapid prototyping derives from the initial use for which the AM technologies were developed. The terms layer-based additive manufacturing and solid freeform fabrication refer to the independence of the geometries that can be produced from the manufacturing process [4]. In this work, the standardized term AM is used.

Over the last three decades, many AM technologies were developed for the production of polymeric, metallic, or ceramic parts. Examples of these techniques include vat photopolymerization (VPP), powder bed fusion (PBF), and material extrusion (MEAM). These three techniques were originally developed for the manufacturing of polymeric parts; however, over the years, they were adapted and used for direct and indirect production of metal and ceramic parts [5].

One of the most commonly used AM technologies for the production of metal parts is PBF. In PBF, a laser or electron beam selectively fuses metal powder or metal powder covered with a binding agent by scanning cross-sectional layers generated from a CAD file of the part on the surface of a powder bed. After one cross-section is scanned, the powder bed is lowered and a new layer of powder is added on top of the previous one; the process repeats until the part is finished [6]. One of the biggest disadvantages of PBF is that it relies on high-power lasers or high-energy electron beams, which can be very costly. In addition, the powder must be free-flowing and, therefore, it requires a specific powder distribution, which adds to the price of the material. Therefore, MEAM shows great promise as a cost-effective alternative since the shaping equipment is orders of magnitude cheaper than PBF [6], and powders with a great range of particle size distributions can be processed. In the most common type of MEAM, the building material is supplied in the form of spooled filaments and, therefore, is also known as fused filament fabrication (FFF). In most FFF machines, the filament is fed into a heating unit with a nozzle using counter-rotating rollers as a feeding system. The extrusion head is controlled to move in the *XY* plane, and, as it moves, material is extruded through the nozzle on a flat platform that moves in the *Z*-direction [5]. However, there are also MEAM systems, where the extrusion system is fixed and the printing platform moves in three axes [7], and systems where the printed head moves in three axes and the building platform is fixed [8].

FFF was first developed for polymeric materials; however, for the fabrication of metal or ceramic parts, a filament made of a polymeric blend filled with a large portion of metal or ceramic particles, known as feedstock, is used. After shaping the filament into what is referred to as the green parts, the binder system is removed from the part by thermal, catalytic, or solvent extraction and then sintered to obtain a final metal or ceramic part. This process was first introduced as FDMet (Fused Deposition of Metals) and FDC (Fused Deposition of Ceramics) in the 1990s [9,10]; however, since developing appropriate binder systems for metal or ceramic powder is not a trivial task and the FFF process was protected by patents [11,12,13], FFF for the production of metal or ceramic parts was neglected until recently. Currently, FFF machines are an open-source technology accessible to almost everyone due to their low price; therefore, the interest in producing metallic or ceramic parts via MEAM was rejuvenated and different companies are offering equipment and materials to produce metal and ceramic specimens indirectly after shaping, debinding, and sintering [14,15,16,17,18]. Moreover, several research groups are investigating different materials [19] such as 316 L steel [20,21,22], 17-4PH steel [23,24,25], Ti6Al4V [26], NdFeB [27], copper [28], zirconia [29], alumina [30,31,32,33], silicon nitride [34,35], lead zirconate titanate [36,37], fused silica [38], tricalcium phosphate [39], hard metals [40,41,42], cermets [41], and multi-material parts combining ceramics and metals [43].

Most materials that can be used in an FFF machine are melt-processable, i.e., they must flow when heated without applying large shear. Conventional FFF machines are basically ram extruders, with the ram being the printing material in the shape of a filament. Therefore, sufficient stiffness is required for the filament to push the material without buckling into the liquefier and through the nozzle. At the same time, the filament must be strong enough to avoid breaking due to cutting by the wheels [44]. In addition to these requirements, the filament should also be flexible enough so that it can be spooled, such that the filament can be easily stored in a compact place and fed in a more or less continuous fashion into the liquefier with very simple mechanisms [27].

Many unfilled or lightly filled thermoplastics fit all these mechanical requirements. However, when the polymers are filled with 45% vol. or more, as is the case with feedstocks, the filaments become very brittle and their melt viscosity increases substantially; therefore, the properties required for ram extrusion are not met with a single-component binder. The only way to achieve the appropriate mechanical properties is to use multicomponent polymeric binder systems. As previously mentioned, finding the right combination of polymers is not a simple task due to the contradictory requirements of FFF and, later on, during debinding. Possible binder compositions for sinterable FFF filaments were discussed previously [19,29].

It was observed that the printing step is crucial in obtaining good mechanical properties of the metal specimens; therefore, it is necessary to find the optimal printing conditions, preferably in a systematic approach. Thus, in this paper, a feedstock formulation consisting of a proprietary binder system [45] and 17-4PH stainless-steel powder was used to shape specimens following a central composite design for optimization, and the tensile properties of the shaped parts were measured after FFF. Several studies are available where the tensile properties of polymeric parts produced by MEAM were investigated. For example, Bayraktar et al. [46], Alafaghani et al. [47], Chacon et al. [48], and Spoerk et al. [49] investigated PLA (Polylactic Acid) parts; Ahn et al. [50], Reddy et al. [51], and Álvarez et al. [52] investigated ABS parts; finally, Spoerk et al. [53] investigated filled polypropylene. However, information of the optimization of the printing conditions to improve the tensile properties of feedstocks to obtain high-quality sintered metallic parts shaped by FFF is not available in the open literature. Such information is crucial in order to extend the applicability of FFF to manufacture uniquely designed or complex load-bearing metallic components that can replace parts made by other manufacturing techniques.

## 2. Materials and Methods

### 2.1. Feedstock Filaments

Feedstock consisted of a multicomponent binder system and filler particles. The binder system consisted of a mixture of a thermoplastic elastomer (Kraiburg TPE GmbH & Co. KG, Waldkraiburg, Germany) with a grafted polyolefin (Byk-Chemie GmbH, Wesel, Germany). The exact binder composition is confidential. The powder was 17-4PH, gas atomized by Sandvik Osprey Ltd., Neath, UK. The particle size data measured by laser diffraction and the chemical composition as given by the producer are shown in Table 1. The particle size distribution is mentioned because it is a factor that greatly determines the FFF processability of the produced feedstocks [54].

Feedstock was prepared in a co-rotating twin-screw extruder (Leistritz ZSE 18 HP-48D, Leistritz Extrusionstechnik GmbH, Nuremberg, Germany). The resulting compound was granulated in a cutting mill (Retsch SM200, Retsch GmbH, Haan, Germany) after it cooled down on air-cooled conveyor belt (Reduction Engineering Scheer, Kent, OH, USA). The feedstock granulates were used to produce filaments in a single-screw extruder (FT-E20T-MP-IS, Dr. Collin GmbH, Ebersberg, Germany) coupled to a Teflon conveyor belt (GAL-25, GEPPERT-BAND GmbH, Jülich, Germany) and a self-developed haul off and winding unit. More details about the process are given elsewhere [24].

### 2.2. Material Extrusion AM Trials

Material extrusion additive manufacturing trials were performed on an Original Prusa i3 MK3 fused filament fabrication machine (Prusa Research, Prague, Czech Republic). Nozzle diameter was 0.4 mm. The printing surface was a replaceable heated steel sheet. The printed parts were dog-bone specimens, as shown in the CAD image in Figure 1.

The software Cura 3D (Ultimaker BV, Utrecht, The Netherlands) was used to prepare the G-code for printing; the following parameters were kept constant: infill density of 100%, rectilinear fill patterns for all layers, fill (raster) angle of 45°, speed of printing of 35 mm/s, extrusion width of first layer of 200%, infill overlap of 15%, and printing surface temperature of 100 °C. The building orientation is as shown in Figure 1, with the broadest dimension against the build platform.

### 2.3. Design of Experiment (DoE)—Central Composite Design

The extruder temperature, the flow rate multiplier, and the layer thickness were varied according to the central composite design for optimization shown in Table 2. Central composite design is a design of first order (2^k^—where k is a number of adjustable parameters) extended with additional trials in the center of the design (mean values of parameters) and axis for each parameter, in order to be able to estimate parameters with models of second order. Central composite design consists of 2^k^ trials on the peak values of the observed parameters, 2^k^ trials in axes of each parameter, and trials at the center of the design (Figure 2).

The desired characteristic of each DoE is the independent estimation of the main factors and their interactions, which is a function of the rotatability of the design. The rotatability of the design is accomplished by adding trials so that all design trials (parameters values) are equally distant from the center of the design (mean values of parameters). In other words, a design is rotatable if it predicts the values of the parameters with the same precision at all points equidistant from the coded origin of the design [56]. A design is rotatable if the following relationship exists [55]:(1)$$\alpha =\sqrt[{4}]{F}$$
where α is the distance of a trial in the design axis from the center of the design (Figure 2), and *F* is the number of parameters.

In the case of central composite design with three adjustable parameters, *F* is equal to 8 (2^3^); therefore, the α value is 1.68. Therefore, design factors in coded form can be presented as −1.68, −1, 0, 1, and 1.68. Total numbers of trials in that case have to be 14 trials on design peaks and at axis and trials in the design center, where the design has repetitions for estimation of the model error.

The determination of the limits of the design of experiment (DoE) was previously discussed in Reference [40]. Five specimens per set of printing conditions were fabricated at a time. A total of 95 parts were produced (Table 2). After the optimal conditions were found, further specimens were reprinted with the optimized conditions to prove the results of the DoE.

### 2.4. Tensile Testing

Tensile testing of 3D-printed specimens was performed on a static material testing machine Shimadzu AGS-X 10 kN fitted with an extensometer (Shimadzu Corporation, Kyoto, Japan) (Figure 3).

Tensile strength (N/mm^2^), maximal tensile force (N), and tensile modulus (N/mm^2^) were used as characterization parameters for the mechanical tensile properties of specimens. Additionally, the mass of the test specimens was measured in order to relate it to the achieved mechanical properties of the tested specimens. Specimens were tested with a deformation speed of 2 mm/min. The results of tensile testing of 3D-printed specimens are presented in Table 3.

### 2.5. Scanning Electron Microscopy (SEM)

For green specimen structure analysis on the fracture surface, the scanning electron microscope Tescan Vega TS 5136 MM (Tescan S.r.o, Brno, Czech Republic) was used (Figure 4).

The microscope had a magnification range of 15 to one million, with a maximal nominal resolution of 3 nm. SEM images were obtained without surface treatment on the specimen cross-section at break.

## 3. Results and Discussion

### 3.1. Fused Filament Fabrication of Specimens

Examples of printed specimens are shown in Figure 5. It can be seen that varying the printing parameters changed the appearance of the 3D-printed specimens.

There was a visible change in the appearance of the printed specimens as the temperature was increased from 210 to 260 °C, with the specimens produced at higher temperature having a slightly smoother surface on the top of the specimen (Figure 5a,b). The factor with the largest effect in the appearance of specimens was the flow rate multiplier since more material was extruded resulting in more overlap between the strands as the multiplier increased from 95% to 127%. For example, having a larger flow multiplier led to a better adhesion to the perimeter layers (Figure 5c,d). The change in the appearance of the parts with the different layer thickness values of the DoE is shown in Figure 5e,f, where specimens with a layer thickness of 0.12 and 0.28 mm are shown.

Figure 6 shows microscopy images of the specimens produced in trial 13 (Figure 6a–c—extrusion temperature 235 °C, flow rate multiplier 127%, and layer thickness 0.2 mm) with the best tensile properties (Table 3) and the filament (Figure 6d).

The microscopy images of the printed specimens show the printed layers (Figure 6a), which were of more or less constant dimensions. In each of the layers and the filament, the steel particles were well distributed and joint together by the polymeric binder (Figure 6b–d). Some pores due to the printing strategy (i.e., places where two perimeters joined or where the infill started and jointed) are visible in Figure 6a,b, and they are marked by dotted lines. This alignment of pores was observed in CT (Computed Tomography) scans of specimens produced by FFF with other highly filled filaments [21], and they can lead to weaker mechanical properties (see Section 3.3).

Further magnification (Figure 6c,d) revealed that the particles were held together by the polymeric binder. Such network morphology is needed in order to be able to perform the steps of debinding and sintering without destroying the shaped parts. It is important to mention that the particle distribution in the printed specimens did not change from that of the filament (Figure 6c,d).

### 3.2. Tensile Properties of 3D-Printed Green Parts

Table 3 presents results of tensile testing 3D-printed specimens. The results are average values of testing five specimens for each DoE trial with standard deviations.

From Table 3, two different DoE trials can be pointed out: trial 3 with the lowest tensile properties, and trial 13 with the highest tensile properties. Figure 7 shows a tensile strength–strain (elongation) diagram for representative specimens from both trials: specimens 3_2 and 13_4. A very big difference can be seen in their tensile behavior, pointing out the importance of selecting the correct printing parameters.

In order to observe trends in the properties and their correlation with adjustable printing parameters, trials 10 (second worst) and 16 (second best) can also be observed and compared to the best and the worst trials. Central composite design with three adjustable parameters was performed at five different levels of adjustable parameters. In coded form, those levels can be presented as −1.68, −1, 0 (center of design), 1, and 1.68 as described in Section 2.4. Table 4 presents a combination of adjustable parameters in coded form for selected trials (3, 10, 13, and 16).

Even before further statistical analysis of the data shown in Table 3 and Table 4, one can conclude that all parameters had a significant effect on the tensile properties (the strongest being the flow rate multiplier) with similar trends; increasing all of the parameters increased the tensile properties. More detailed analysis is made in Section 3.3.

### 3.3. Statistical Analysis of Tensile Testing Results

For statistical analysis of the tensile testing results, the software Design Expert (Stat-Ease Inc., Minneapolis, MN, USA) was used. Central composite design, as a part of response surface methodology, is a statistical technique useful for developing, improving, and optimizing processes. In statistical analysis of the testing results, two tensile properties were examined (tensile strength and tensile modulus). Analysis of variance (ANOVA) and response surface models were used in order to determine significance of adjustable printing parameters and their interactions on the observed tensile properties. Statistical analysis was also used to determine a mathematical function, which can be used for the prediction of future results for the observed tensile properties.

#### 3.3.1. Statistical Analysis of Tensile Strength

Results of ANOVA for tensile strength testing are shown in Table 5. The mathematical model of the response surface (quadratic model) for tensile strength (TS) can be presented in the form of Equation (2), where all factors are in coded form as specified in Table 5. The equation in terms of coded factors can be used to make predictions about the response for given levels of each factor. By default, the high levels of the factors are coded as +1 and the low levels are coded as −1. The coded equation is useful for identifying the relative impact of the factors by comparing the factor coefficients.
*TS* = 6.807 + 0.298⋅A + 1.333⋅B + 0.467⋅C − 0.013⋅AB − 0.028⋅AC − 0.027⋅BC − 0.111⋅A^2^ − 0.275⋅B^2^ + 0.025⋅C^2^.(2)

From Table 5, it can be concluded that all parameters (i.e., flow rate multiplier, layer thickness, and extrusion temperature) had a strong and significant impact on the tensile strength; increasing the value of these parameters resulted in an increased tensile strength. The flow rate multiplier increased the amount of material being pushed through the nozzle (i.e., the volumetric flow rate), resulting in heavier specimens (Table 3), in which the deposited strands were better connected and overlapping; therefore, the amount of voids between strands was reduced. It was determined that voids are the main factor leading to reduced mechanical properties in FFF parts [57]; therefore, reducing voids is very important to improve the mechanical performance. This is especially visible in trials 1, 13, and 16, where this factor was set to higher values (120% and 127%) (Figure 8). The results of the DoE are in agreement with studies performed with unfilled PLA and PLA filled with carbon nanotubes, in which the tensile strength increased upon increasing the volume flow rate [49,58].

The flow rate multiplier has a limitation on higher levels, because pushing excessive material through the extruder nozzle can result in nozzle clogging, which can lead to void creation in the fabricated specimens [59]; for this reason, it was limited to 127% in the DoE.

Increasing the layer thickness also increased the tensile strength. This was also observed in unfilled PLA specimens, particularly when the raster angle was 45° and a flat printing orientation was used [60,61], similar to the conditions used in the present DoE. Increasing the layer thickness is equivalent to decreasing the number of layers and interlayer contacts. The interlayer contacts could be the weakest points of the printed specimens depending on the printing conditions, since there is an incomplete diffusion of polymer chains between adjacent strands, a reduced cross-section due to the introduction of voids, and fracture mechanic-type stress concentrations [43]. Therefore, decreasing the number of interfaces by increasing the layer thickness can increase the strength of the specimens [35]. Layer thickness is also limited due to the 3D printer design; most of the printers for material extrusion can achieve layer thicknesses between 0.1 and 0.3 mm [60], which are within the limits of the DoE used. It is important to mention that, if the building orientation, the raster angle, and the raster width are changed, the trend of the tensile strength, as a function of layer thickness, can be reversed, as reported previously for other materials [49,60,61].

The results of the tensile strength dependence on the extrusion temperature are also in agreement with the results obtained for PLA [49,62,63,64], in which an increase in extrusion temperature led to higher tensile strengths. The increase in tensile properties can be explained by the improved fluidity (decreasing viscosity) of the material, which decreases flow resistance as it passes through the nozzle. With lower extrusion temperature, the lack of material deposition results in bigger air gaps, which reduce the cross-section of the specimens. That can be confirmed with results of specimen mass analysis (Table 2 and Table 3), where specimens printed at higher temperatures have higher masses even without the interaction with flow rate multiplier, as the parameter with the strongest effect on specimen mass and tensile properties. Extrusion at lower temperatures also results in a reduced bonding between two different layers that leads to lower tensile properties. It has to be pointed out that, in this research, only the polymer component (45% vol.) was melted during the FFF process; therefore, the effect of extrusion temperature on tensile properties was not as strong as in case of FFF of pure polymer material [46,47,48,49,50,51,52,53].

The square of the flow rate multiplier (B^2^) also has a significant impact on the tensile strength of specimens. However, the effect is much lower compared to the influence of all parameters independently.

The influence of the extrusion temperature, flow rate multiplier, and layer thickness on the tensile strength of fabricated parts can be presented in the form of two-dimensional (2D) graphs (Figure 9, Figure 10 and Figure 11). While presenting individual parameter, the other two are maintained at mean values.

From Figure 9, Figure 10 and Figure 11, it can be concluded that increasing all three observed parameters individually led to an increase in tensile strength. Influence of the flow rate multiplier (Figure 10) was the most significant parameter here, as seen by the steeper slope of the curve.

Figure 12, Figure 13 and Figure 14 show the influence of two parameters on tensile strength simultaneously in the form of 3D graphs, while keeping the third parameter at the maximum value.

From Figure 12, Figure 13 and Figure 14, it can be concluded that the flow rate multiplier had the most significant effect on tensile strength. On Figure 12 and Figure 14, a significant increase in tensile strength can be seen upon increasing the flow rate multiplier in interaction with the extrusion temperature and layer thickness. The interaction of the other two parameters had a slight influence (Figure 13) on tensile strength; thus, the plotted surface area appeared almost flat.

#### 3.3.2. Statistical Analysis of Tensile Modulus

Results of ANOVA for the tensile modulus testing of printed specimens are shown in Table 6.

The mathematical model of the quadratic response surface for tensile modulus (TM) can be presented in the form of Equation (3), where all factors are in coded form according to Table 6.
TM = 179.18 + 5.12⋅A + 29.69⋅B + 17.47⋅C + 2.54⋅AB + 1.90⋅AC + 0.408⋅BC − 0.931⋅A^2^ − 4.98⋅B^2^ − 3.12⋅C^2^.(3)

Here, flow rate multiplier and layer thickness are significant parameters. Figure 15, Figure 16 and Figure 17 show the individual effect of all three parameters on tensile modulus, while keeping the other two parameters at mean values.

As presented in Figure 15, Figure 16 and Figure 17, increasing both flow rate multiplier and layer thickness led to an increase in tensile modulus (Figure 18), with a stronger effect of the flow rate multiplier (Figure 16).

As the tensile modulus describes material behavior in the elastic area and presents a measure of the stiffness of the specimens, it is expected that a specimen printed with higher flow rate multiplier and with larger layer thickness would result in higher density and, as a result, stiffer specimens. Extrusion temperature also had the same effect (Figure 15); however, in this case, it was not significant.

### 3.4. Statistical Model for Optimization

In order to set the FFF parameters for the production of specimens with optimal properties, the criteria for optimization have to be determined (Table 7). Here, complex optimization based on multiple simultaneous criteria was used. The limits of the expected tensile properties, shown in Table 7, were set according to the results obtained from the previous DoE analyzed in Section 3.3.

The mathematical method for finding the optimum of the observed response surface was the desirability function. The desirability function ranges from 0 to 1, where 1 is the goal of the optimization. The numerical results of the complex optimization with the highest desirability function are shown in Table 8.

A graphical representation of optimized adjustable parameters is shown in Figure 19, Figure 20 and Figure 21.

All three observed FFF parameters (extrusion temperature, flow rate multiplier, and layer thickness) have to be maximized within the previously determined processing window in order to maximize the tensile properties of the green specimens. From Figure 19, Figure 20 and Figure 21, it is clear that increasing the flow rate multiplier resulted in the highest increase of all observed tensile properties, while extrusion temperature had the lowest effect on the observed properties. As already mentioned, the extrusion temperature only had a strong effect on the polymeric components (i.e., 45% vol. of the feedstock); therefore, its effects on the mechanical properties were generally at a lower level compared to the properties of pure polymeric FFF specimens [49].

For confirming the accuracy of the estimated values of tensile properties, a new set of green specimens was reprinted with the parameters defined in Table 8. Figure 22 presents a reprinted specimen with the optimized FFF parameters.

Table 9 presents the results of tensile testing, as well as variance from the estimated values. Figure 23 presents the tensile strength–strain diagram for the reprinted specimens (opt_1, opt_2 and opt_3), as well as the diagram of the average value (opt_av).

All the values of the tensile properties from Table 9 show good correlation with the estimated values obtained from the DoE optimization (Table 8), which confirms that DoE can be used for the prediction of properties of FFF printed specimens with satisfactory accuracy. Upon using the model obtained by the DoE to set the printing parameters, the tensile strength was improved by 17%, the maximum force was improved by 6%, and the tensile modulus was improved by 23% as compared to trial 13, which originally gave the best results.

## 4. Conclusions and Future Work

In the paper, the effects of three FFF parameters (extrusion temperature, flow rate multiplier, and layer thickness) on the tensile properties (tensile strength and tensile modulus) of the stainless-steel (17-4PH)–polymer composite dog-bone specimens were presented. For analyzing the effects, central composite design with three parameters was used. Effects of all selected FFF parameters showed the same trend; increasing the mentioned parameters resulted in increased tensile properties of the green specimens. Flow rate multiplier had the strongest effect on tensile properties since more material was pushed through the nozzle, resulting in more compact specimens in which the deposited strands were better connected and overlapping, leading to a reduction of voids between strands. The interlayer contacts could be the weakest points of the printed specimens depending on the printing conditions, since there was an incomplete diffusion of polymer chains between adjacent strands, a reduced cross-section due to the introduction of voids, and fracture mechanic-type stress concentrations. Therefore, decreasing the number of layers (increasing the layer thickness) resulted in green specimens with larger tensile properties. A lower extrusion temperature interfered with material deposition, resulting in more air gaps and reduced minimal cross-section of the specimens and, therefore, lower tensile properties. Extrusion at lower temperatures also negatively affected the bond between two different layers, which led to lower tensile properties. In this research, only the polymer components (45% vol.) were melted during FFF process; therefore, the effect of extrusion temperature on tensile properties was not as strong as in the case of FFF of pure polymer materials.

The results of this analysis can be used in future research for comparison with properties of final parts obtained after green parts undergo debinding and sintering. This comparison will show if the same trend between tensile properties obtained after FFF and after sintering are present, and if the level of the effect of tensile properties obtained after the FFF process holds after the specimens are sintered.

## Figures and Tables

**Figure 1 materials-13-00774-f001:**
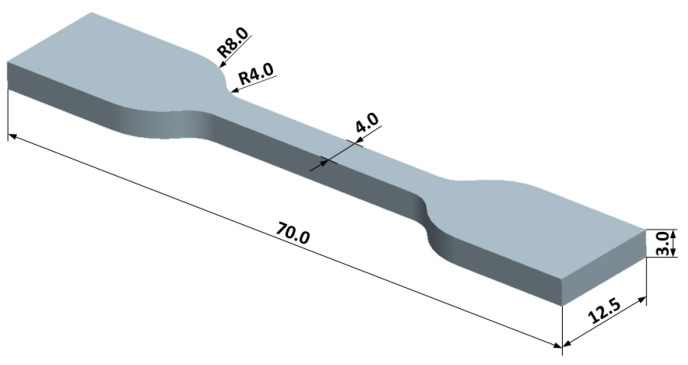
Dog-bone specimen printed during printing trials (all dimensions are in mm).

**Figure 2 materials-13-00774-f002:**
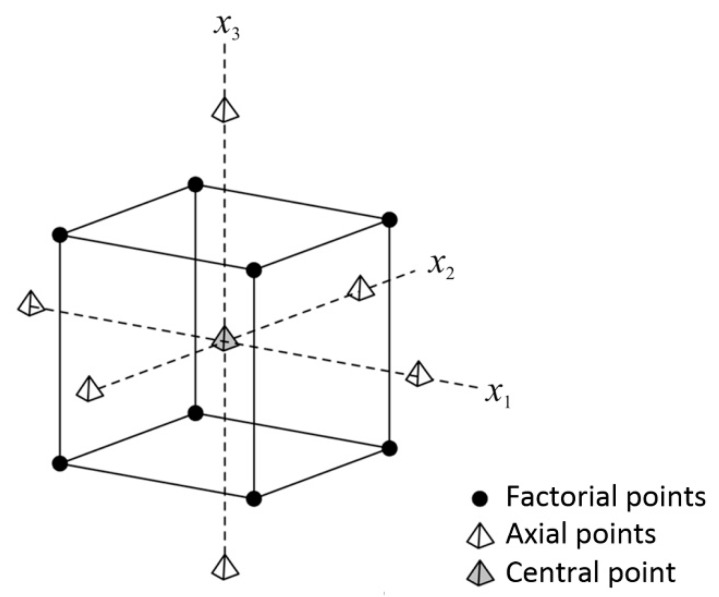
Model of central composite design with three factors [55].

**Figure 3 materials-13-00774-f003:**
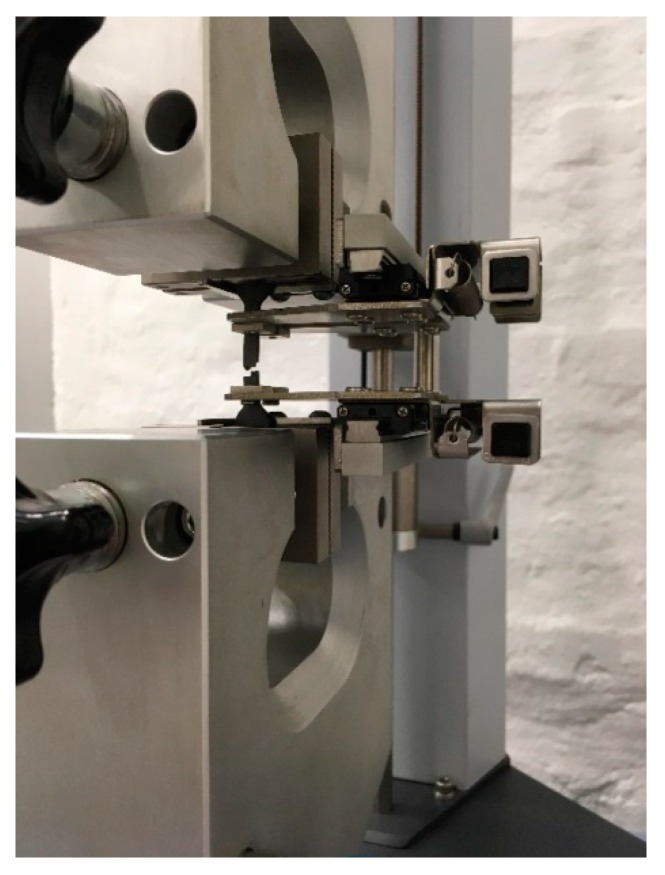
Static material testing machine Shimadzu AGS-X 10 kN with tested specimen.

**Figure 4 materials-13-00774-f004:**
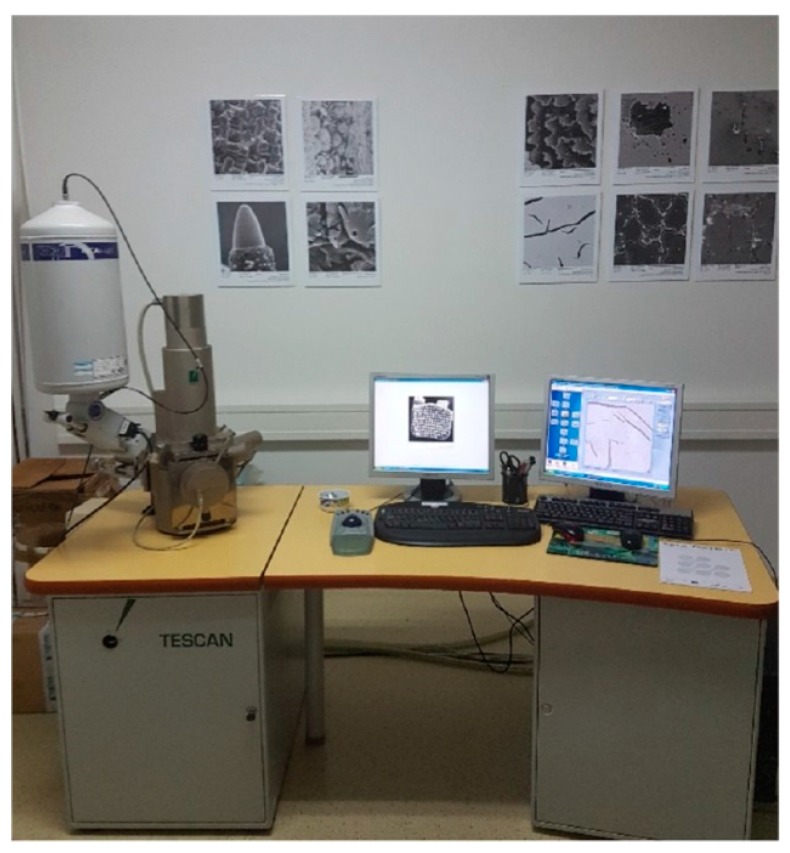
Scanning electron microscope Tescan Vega TS 5136 MM.

**Figure 5 materials-13-00774-f005:**
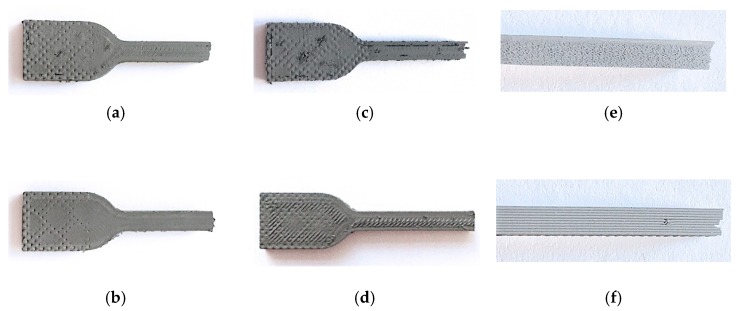
Top and side views of specimens printed at different conditions of the design of experiment (DoE) after tensile testing: (**a**) trial 7 (temperature 210 °C); (**b**) trial 5 (temperature 260 °C); (**c**) trial 10 (flow rate multiplier 95%); (**d**) trial 13 (flow rate multiplier 127%); (**e**) trial 2 (layer thickness. 0.12 mm); (**f**) trial 6 (layer thickness. 0.28 mm).

**Figure 6 materials-13-00774-f006:**
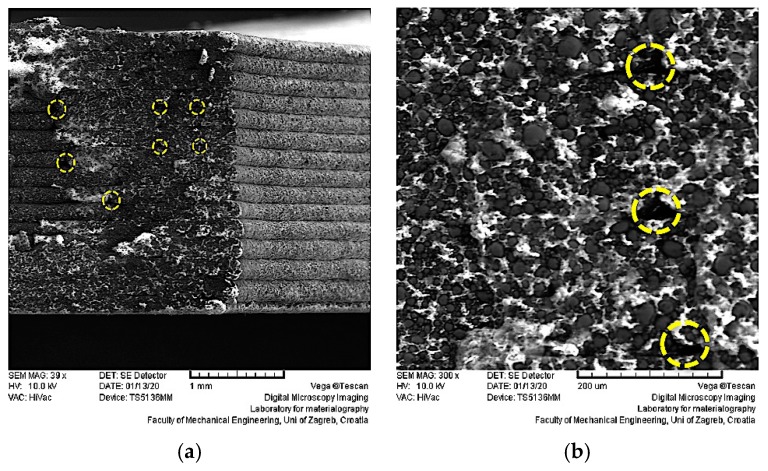
Morphology of three-dimensional (3D) printed parts investigated by scanning electron microscopy at different magnifications: (**a**) 39×; (**b**) 300×; (**c**) 2300×; (**d**) 2400×.

**Figure 7 materials-13-00774-f007:**
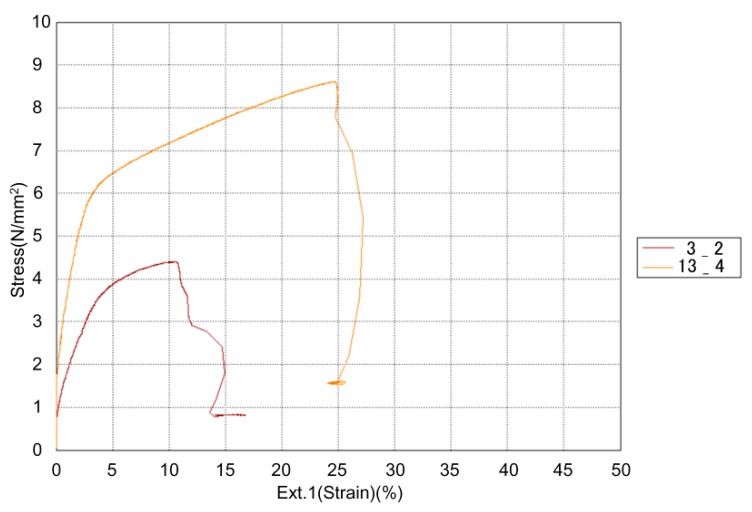
Tensile strength–strain diagram for strongest and weakest specimens: 3_2 and 13_4.

**Figure 8 materials-13-00774-f008:**
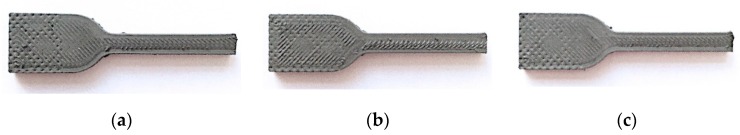
Top view of specimens from trials with highest values of flow rate multiplier: (**a**) trial 1; (**b**) trial 13; (**c**) trial 16.

**Figure 9 materials-13-00774-f009:**
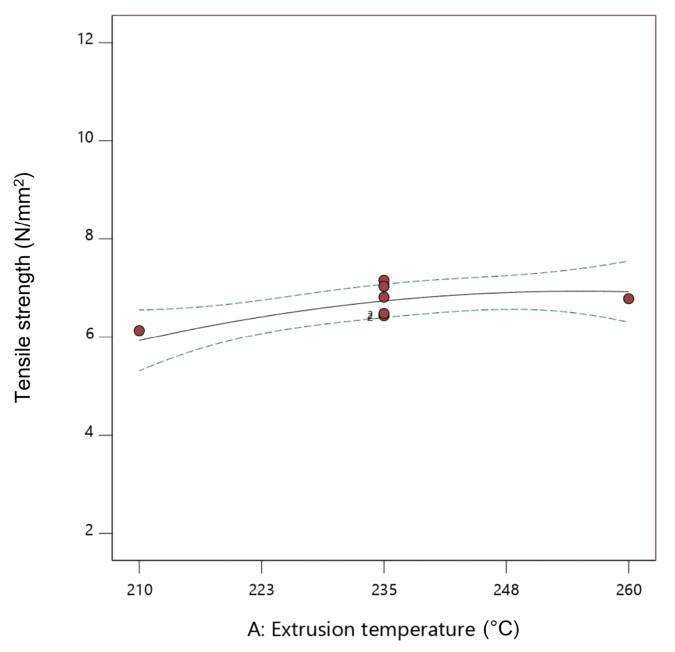
Two-dimensional (2D) graph of the influence of extrusion temperature on tensile strength (flow rate multiplier 110%, layer thickness 0.20 mm).

**Figure 10 materials-13-00774-f010:**
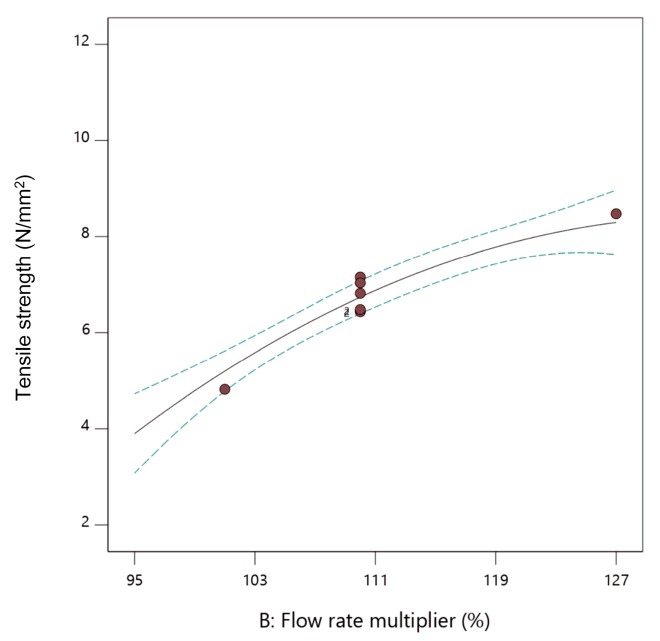
The 2D graph of the influence of flow rate multiplier on tensile strength (extrusion temperature 235 °C, layer thickness 0.20 mm).

**Figure 11 materials-13-00774-f011:**
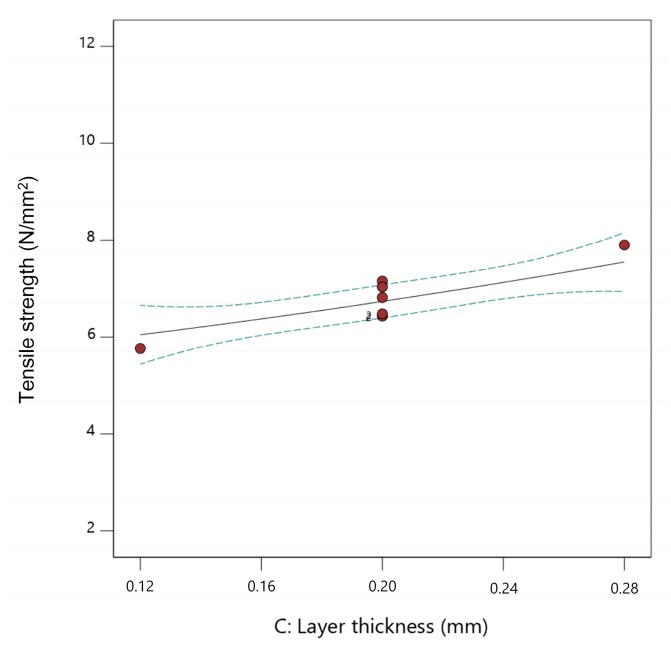
The 2D graph of the influence of layer thickness on tensile strength (extrusion temperature 235 °C, flow rate multiplier 110%).

**Figure 12 materials-13-00774-f012:**
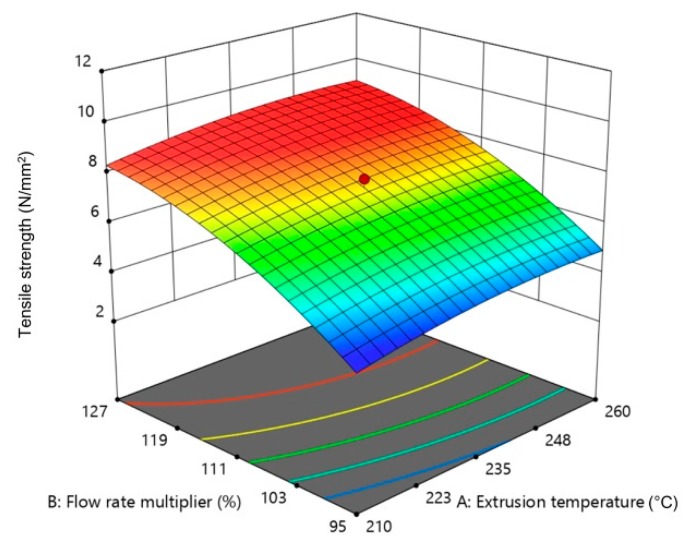
The 3D graph of the simultaneous influence of extrusion temperature and flow rate multiplier on tensile strength (layer thickness 0.28 mm).

**Figure 13 materials-13-00774-f013:**
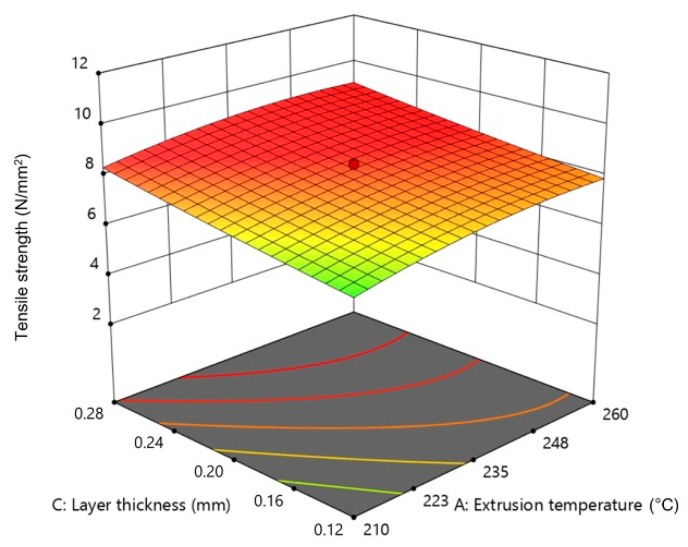
The 3D graph of the simultaneous influence of extrusion temperature and layer thickness on tensile strength (flow rate multiplier 127%).

**Figure 14 materials-13-00774-f014:**
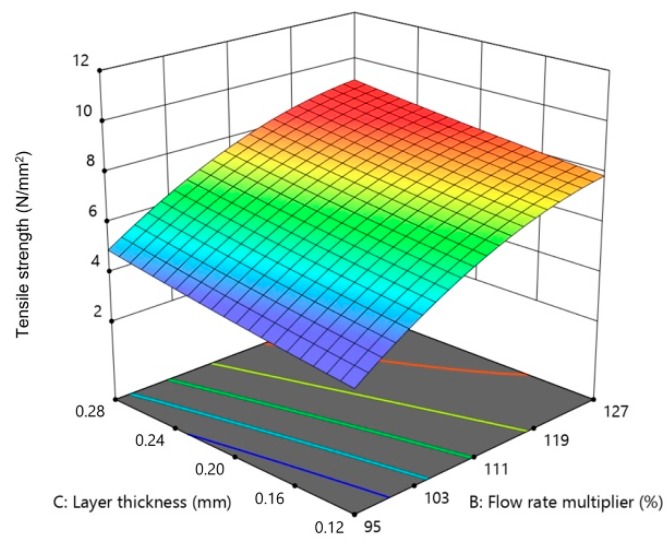
The 3D graph of the simultaneous influence of flow rate multiplier and layer thickness on tensile strength (extrusion temperature 260 °C).

**Figure 15 materials-13-00774-f015:**
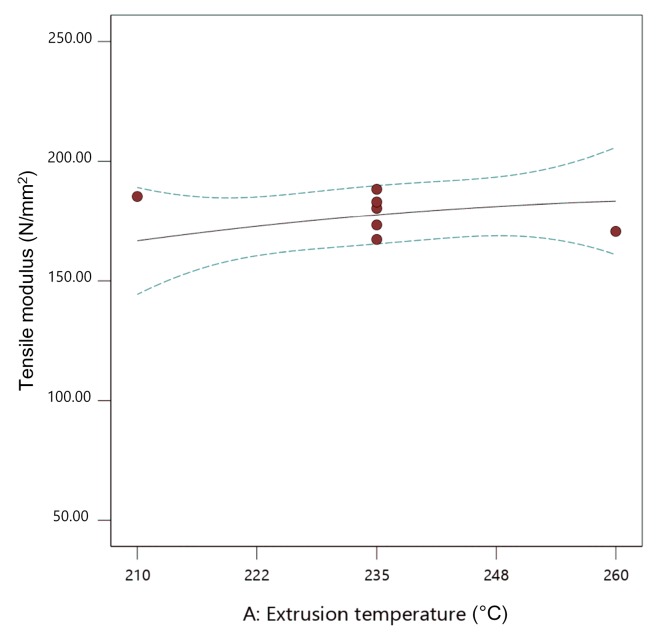
The 2D graph of the influence of extrusion temperature on the tensile modulus (flow rate multiplier 110%, layer thickness 0.20 mm).

**Figure 16 materials-13-00774-f016:**
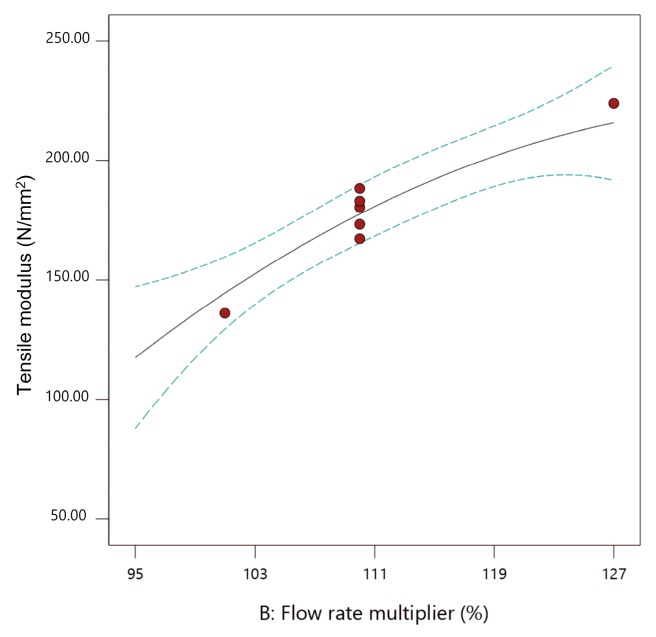
The 2D graph of the influence of flow rate multiplier on the tensile modulus (extrusion temperature 235 °C, layer thickness 0.20 mm).

**Figure 17 materials-13-00774-f017:**
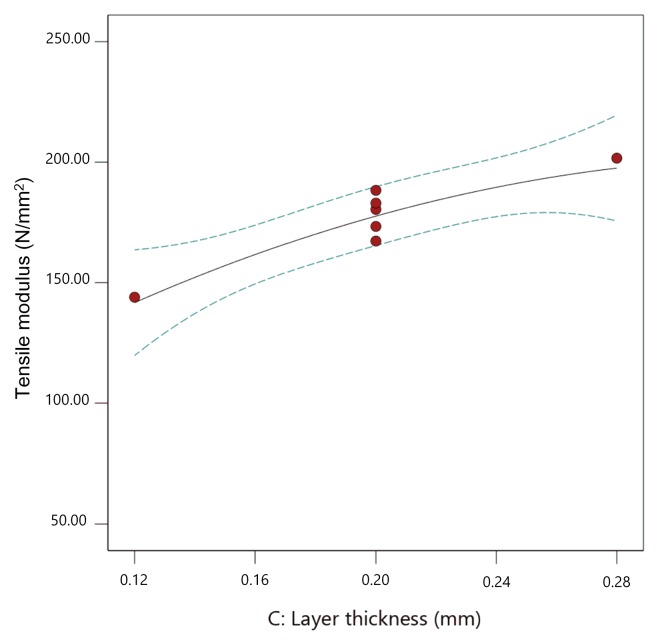
The 2D graph of the influence of layer thickness on the tensile modulus (flow rate multiplier 110%, extrusion temperature 235 °C).

**Figure 18 materials-13-00774-f018:**
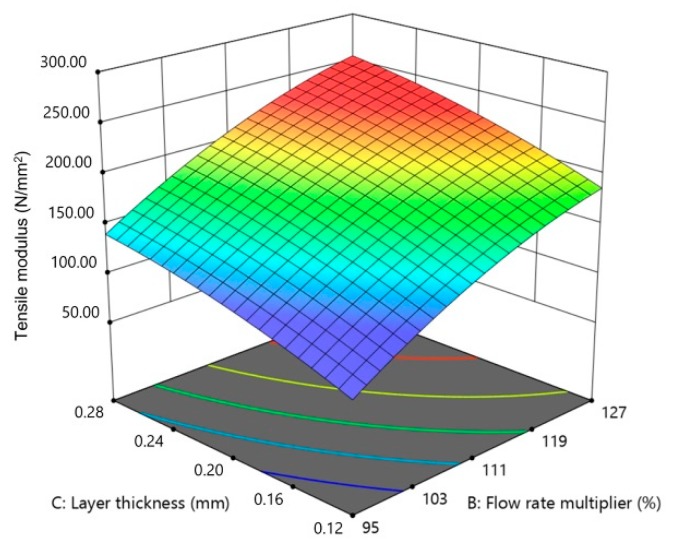
Simultaneous effect of flow rate multiplier and layer thickness on the tensile modulus (extrusion temperature 260 °C).

**Figure 19 materials-13-00774-f019:**
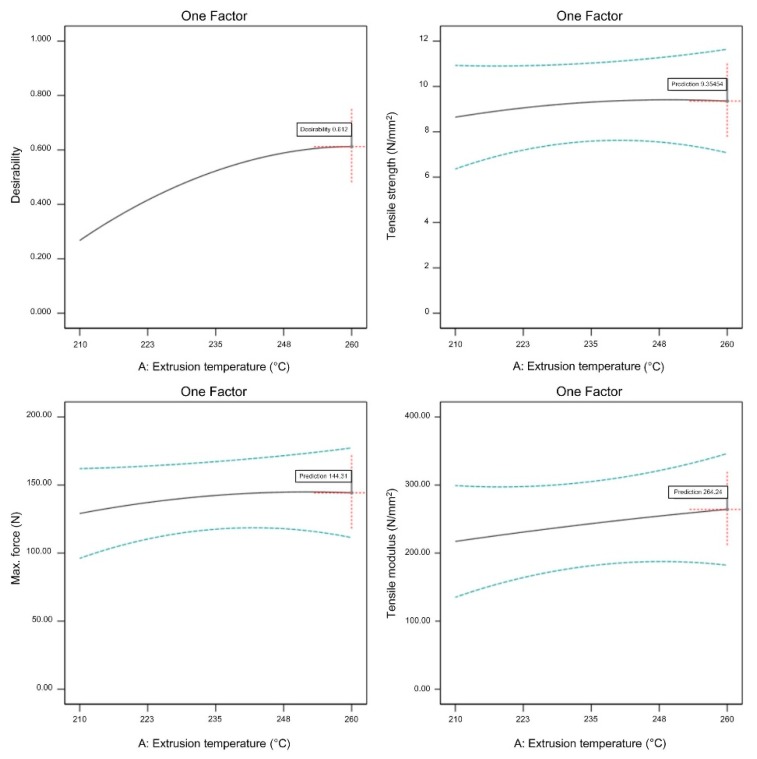
Optimized extrusion temperature (flow rate multiplier 130%, layer thickness 0.3 mm).

**Figure 20 materials-13-00774-f020:**
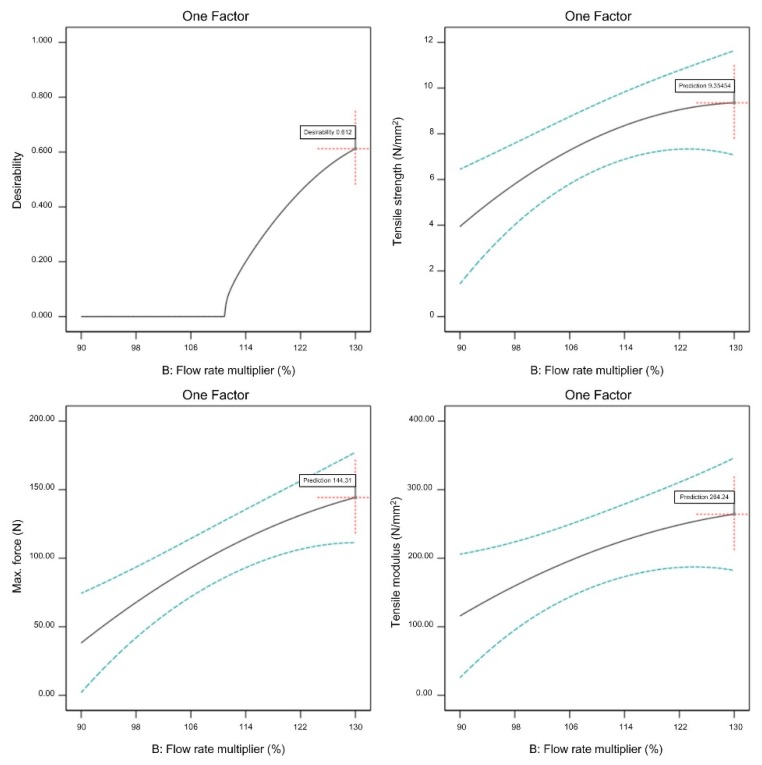
Optimized flow rate multiplier (extrusion temperature 260 °C, layer thickness 0.3 mm).

**Figure 21 materials-13-00774-f021:**
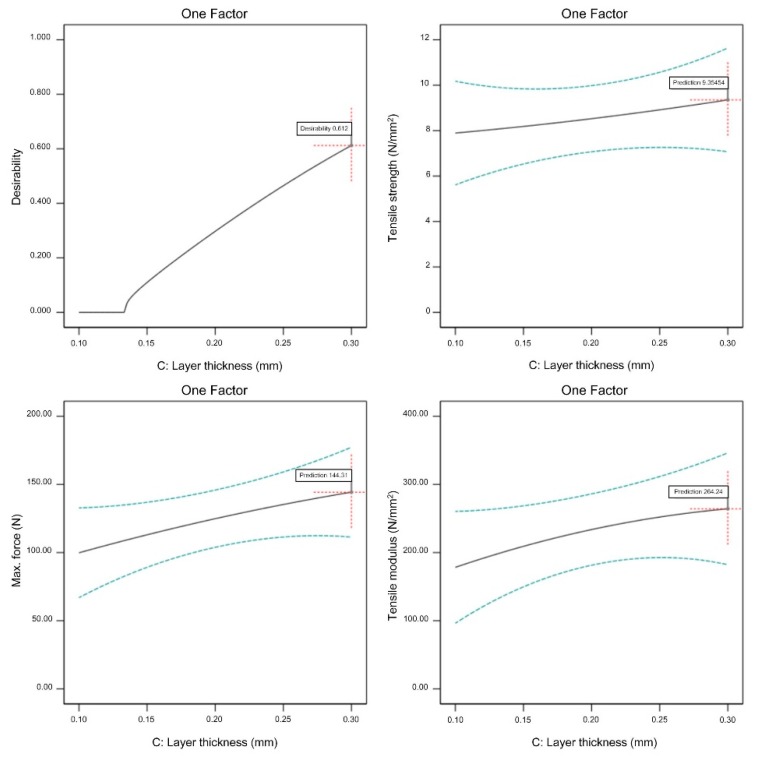
Optimized layer thickness (extrusion temperature 260 °C, flow rate multiplier 130%).

**Figure 22 materials-13-00774-f022:**
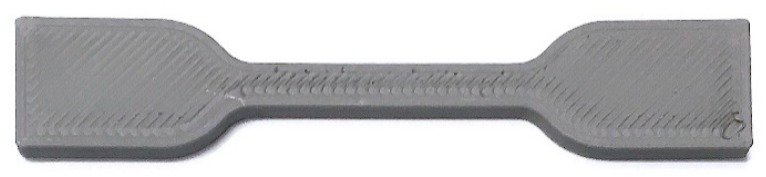
Specimen reprinted with optimized fused filament fabrication (FFF) parameters (extrusion temperature 260 °C, flow rate multiplier 130%, layer thickness 0.30 mm).

**Figure 23 materials-13-00774-f023:**
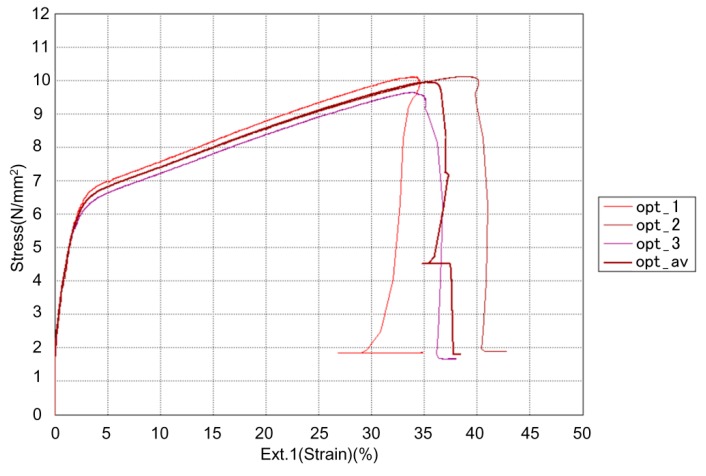
Tensile strength–strain diagram for reprinted (optimized) specimens.

**Table 1 materials-13-00774-t001:** Particle size data of 17-4PH stainless-steel powder.

Particle Size	Distribution
D10 (μm)	4.2
D50 (μm)	12.3
D90 (μm)	28.2

**Table 2 materials-13-00774-t002:** Central composite design for optimization (19 trials; five repetitions in the center).

Trial Number	Factor 1: Extrusion Temperature (°C)	Factor 2: Flow Rate Multiplier (°C)	Factor 3: Layer Thickness (mm)
1	220	120	0.25
2	235	110	0.12
3	220	101	0.15
4	250	101	0.15
5	260	110	0.20
6	235	110	0.28
7	210	110	0.20
8	220	120	0.15
9	250	101	0.25
10	235	95	0.20
11	235	110	0.20
12	235	110	0.20
13	235	127	0.20
14	235	110	0.20
15	235	110	0.20
16	250	120	0.25
17	235	110	0.20
18	250	120	0.15
19	220	101	0.25

**Table 3 materials-13-00774-t003:** Results of tensile testing of green parts (averages with standard deviations (± SD)).

Trial Number	Tensile Strength ± SD (N/mm^2^)	Maximal Force ± SD (N)	Tensile Modulus ± SD (N/mm^2^)	Mass (g)
1	7.67 ± 0.51	109.12 ± 6.77	194.70 ± 29.01	6.57
2	5.76 ± 0.15	74.32 ± 1.88	143.97 ± 9.55	5.10
3	4.40 ± 0.20	56.54 ± 2.30	114.75 ± 9.78	4.80
4	5.23 ± 0.23	62.36 ± 2.90	134.16 ± 14.94	4.90
5	6.78 ± 0.37	90.63 ± 5.19	170.62 ± 29.26	5.70
6	7.90 ± 0.49	106.61 ± 7.41	201.65 ± 17.02	6.30
7	6.13 ± 0.22	82.17 ± 3.95	185.30 ± 22.08	5.70
8	6.97 ± 0.26	91.14 ± 3.90	167.77 ± 25.42	5.70
9	5.89 ± 0.33	74.93 ± 4.12	166.99 ± 30.59	5.70
10	4.82 ± 0.25	62.58 ± 2.98	136.08 ± 17.87	5.10
11	7.15 ± 0.31	95.37 ± 3.65	173.34 ± 14.94	5.80
12	7.03 ± 0.33	93.60 ± 5.25	180.41 ± 17.55	5.80
13	8.47 ± 0.13	124.51 ± 2.31	223.92 ± 12.65	6.70
14	6.82 ± 0.23	94.14 ± 3.48	188.30 ± 17.11	5.80
15	6.44 ± 0.41	84.55 ± 5.60	183.01 ± 25.80	5.67
16	8.32 ± 0.48	116.80 ± 6.95	230.50 ± 21.69	6.70
17	6.48 ± 0.18	86.79 ± 2.49	167.25 ± 10.40	5.80
18	7.73 ± 0.24	101.81 ± 2.74	187.32 ± 16.04	5.80
19	5.18 ± 0.37	65.35 ± 4.75	148.62 ± 11.14	5.43

**Table 4 materials-13-00774-t004:** Adjustable parameters and their values in coded form (trials 3, 10, 13, and 16).

Trial Number	Factor 1: Extrusion Temperature	Factor 2: Flow Rate Multiplier	Factor 3: Layer Thickness
*The worst trials (combinations of parameters in coded form)*			
3	−1	−1	−1
10	0	−1.68	0
*The best trials (combinations of parameters in coded form)*			
13	0	1.68	0
16	1	1	1

**Table 5 materials-13-00774-t005:** ANOVA for tensile strength testing (quadratic model, DoF—degree of freedom).

Source	Sum of Squares	DoF	Mean Square	F-Value	*p*-Value	Remark
Model	23.56	9	2.62	21.49	<0.0001	Significant
A—Extrusion temperature	1.20	1	1.20	9.84	0.0120	Significant
B—Flow rate multiplier	18.56	1	18.56	152.31	<0.0001	Significant
C—Layer thickness	2.86	1	2.86	23.51	0.0009	Significant
AB	0.0015	1	0.0015	0.0119	0.9154	
AC	0.0064	1	0.0064	0.0524	0.8241	
BC	0.0058	1	0.0058	0.0477	0.8320	
A²	0.1682	1	0.1682	1.38	0.2701	
B²	0.6866	1	0.6866	5.63	0.0417	Significant
C²	0.0072	1	0.0072	0.0592	0.8132	
Residual	1.10	9	0.1219			
Lack of fit	0.6840	5	0.1368	1.33	0.4038	Not significant
Pure Error	0.4127	4	0.1032			
Corrected Total	24.66	18				

**Table 6 materials-13-00774-t006:** ANOVA for tensile modulus testing (quadratic model, DoF—degree of freedom).

Source	Sum of Squares	DoF	Mean Square	F-Value	*p*-Value	Remark
Model	14,316.59	9	1590.73	10.11	0.0010	Significant
A—Extrusion temperature	355.08	1	355.08	2.26	0.1674	
B—Flow rate multiplier	9204.05	1	9204.05	58.47	<0.0001	Significant
C—Layer thickness	4001.95	1	4001.95	25.42	0.0007	Significant
AB	51.60	1	51.60	0.3278	0.5810	
AC	28.91	1	28.91	0.1837	0.6783	
BC	1.33	1	1.33	0.0085	0.9288	
A²	11.75	1	11.75	0.0747	0.7908	
B²	225.21	1	225.21	1.43	0.2622	
C²	115.74	1	115.74	0.7352	0.4134	
*Residual*	1416.72	9	157.41			
Lack of fit	1143.52	5	228.70	3.35	0.1325	Not significant
Pure Error	273.20	4	68.30			
Corrected Total	15,733.31	18				

**Table 7 materials-13-00774-t007:** Optimization criteria.

Parameter/Property	Goal	Lower Limit	Upper Limit	Importance
A: Extrusion temperature	Is in range	200 °C	260 °C	3
B: Flow rate multiplier	Is in range	90%	130%	3
C: Layer thickness	Is in range	0.1 mm	0.3 mm	3
Tensile strength	Maximize	8 N/mm^2^	12 N/mm^2^	5
Maximum tensile force	Maximize	100 N	160 N	5
Tensile modulus	Maximize	200 N/mm^2^	270 N/mm^2^	5

**Table 8 materials-13-00774-t008:** Numerical results of optimization (highest desirability function).

Parameter/Property	Goal	Unit
A: Extrusion temperature	260	°C
B: Flow rate multiplier	130	%
C: Layer thickness	0.3	mm
Tensile strength	9.36	N/mm^2^
Maximum tensile force	144.31	N
Tensile modulus	264.24	N/mm^2^

**Table 9 materials-13-00774-t009:** Tensile properties of the reprinted green specimens.

Property	Value ± sd	Unit	Variance from Estimated
Tensile strength	9.95 ± 0.27	N/mm^2^	+6.3
Maximum tensile force	132.27 ± 2.86	N	−8.3
Tensile modulus	275.14 ± 6.22	N/mm^2^	+4.1

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
