# Peer review of "Optimization of the 3D Printing Parameters for Tensile Properties of Specimens Produced by Fused Filament Fabrication of 17-4PH Stainless Steel"

_materials, 2020, doi:10.3390/ma13030774_

Round 1

Reviewer 1 Report

The manuscript under the title: Optimisation of the 3D printing parameters for tensile properties of specimens produced by fused filament fabrication of 17-4PH stainless steel”  is suitable for the Materials journal. The authors work on up-to-date topic connected with 3D printing technology. The authors join computer modelling with the experimental research work. The organization of the article is appropriate. Overall, the paper is well prepared. Nevertheless, the article required some improvements:

- the abstract – please add the main aim of article / research

- the introduction part – please reformulate the sentence: “AM is the term standardize by ISO and ASTM [2]; however other names for AM include: 3D printing, solid freeform fabrication (SFF) and rapid prototyping (RP) [3].” Please add some information about the equivalency or not this terms – in the literature it is quite important discussion in this area between some authors.

- the introduction part – please verify the references [18–29], be more accurate about citations (!)

Author Response

Reviewer 1.

The manuscript under the title: Optimisation of the 3D printing parameters for tensile properties of specimens produced by fused filament fabrication of 17-4PH stainless steel”  is suitable for the Materials journal. The authors work on up-to-date topic connected with 3D printing technology. The authors join computer modelling with the experimental research work. The organization of the article is appropriate. Overall, the paper is well prepared. Nevertheless, the article required some improvements:

The authors thank the review for the positive assessment of the work. We have taken into account the reviewer’s comments and a response to each one of them is shown below.

- the abstract – please add the main aim of article / research

We agree with the reviewer that the main goal of the article should be added to the abstract. The abstract has been modified as follows to incorporate the goal of the article (Highlighted in yellow):

Fused filament fabrication (FFF) combined with debinding and sintering could be an economical process for 3D printing of metal parts. The shaping step by FFF is crucial, since printing defects cannot be corrected in the subsequent steps; therefore this paper aims at showing a methodology for optimising the printing conditions in order to maximize the mechanical performance of specimens. For that purpose, in this paper compounding, filament making and FFF processing of feedstock material with 55 % vol of 17-4PH stainless steel powder in a multicomponent binder system is presented. The experimental part of the paper encompass Central Composite Design for Optimisation of the most significant 3D printing parameters (extrusion temperature, flow rate multiplier and layer thickness) to obtain maximum tensile strength of the 3D printed specimens. Here only green specimens were examined in order to be able to determine the optimal parameters for 3D printing. The results show that the factor that has the biggest influence on the tensile properties was flow rate multiplier, followed by the layer thickness and finally the extrusion temperature. Maximising all three parameters leads to the highest tensile properties of the green parts.

- the introduction part – please reformulate the sentence: “AM is the term standardize by ISO and ASTM [2]; however other names for AM include: 3D printing, solid freeform fabrication (SFF) and rapid prototyping (RP) [3].” Please add some information about the equivalency or not these terms – in the literature it is quite important discussion in this area between some authors.

The authors appreciate the insightful comment and agree that this debate should be mentioned in the introduction to inform readers about standardization efforts in the field of AM. Therefore we have added to the introduction the following sentences:

Additive manufacturing (AM) comprises a group of technologies used to build physical parts by adding material in a layer-by-layer fashion from a computer aided design (CAD) file; as opposed to subtractive manufacturing methods, such as machining [1]. AM is the term standardized by ISO and ASTM [2]; however other common names for AM include: 3D printing, layer-based manufacturing, solid freeform fabrication (SFF) and rapid prototyping (RP) [3,4]. Each of these terms highlights a distinct feature of these processes, for example 3D printing is a common name used by the general public, whereas the term rapid prototyping derives from the initial use for which the AM technologies were developed. The terms layer-based additive manufacturing and solid freeform fabrication refer to the independence of the geometries that can be produced from the manufacturing process [3]. In this work, the standardized term AM will be used.

- the introduction part – please verify the references [18–29], be more accurate about citations (!)

Thank you for your comment. We agree that the citations should be more specific; therefore we had added the citation number after each type of material. Also we have updated the references to be more specific in the field of ceramics and multi-material components. Thus, the paragraph of the introduction reads as follows:

And several research groups are investigating different materials [19] such as 316 L steel [20–22], 17-4PH steel [23–25], Ti6Al4V [26], NdFeB [27], copper [28], zirconia [29], alumina [30–33], silicon nitride [34,35], lead zirconate titanate [36,37], fused silica [38], tricalcium phosphate [39], hard metals [40–42], cermets [42] and multi-material parts combining ceramics and metals [43].

Reviewer 2 Report

The main aim of the study was to investigate the effect of extrusion temperature, flow rate multiplier and layer thickness on tensile properties of FFF parts. In which Flow rate multiplier has the strongest effect on tensile properties since more material is being pushed through the nozzle during the fabrication process.

The article is well written and the details are enough to give proof of presented idea, however, there are some minor changes which should be made in order to improve the clarity and quality of presented work:

Section 2.2: Some dimensions (Fillet radius) of dog-bone specimen are missing. Section 2.3-line(141): “rotability” is wrong word if not then what is the meaning of this word. Section 2.3-line(143): Explain “rotatable” word for better understanding of readers. The figure name is not centered with figures (Fig. 1,2-4). The quality of graphs needs improvement (Fig. 7,19). Section 3.1 is well written, however some sections is lacking the flow of sentences like Section 2.4. revise and modify for the improvement of paper quality.

Author Response

Reviewer 2.

The main aim of the study was to investigate the effect of extrusion temperature, flow rate multiplier and layer thickness on tensile properties of FFF parts. In which Flow rate multiplier has the strongest effect on tensile properties since more material is being pushed through the nozzle during the fabrication process.

The article is well written and the details are enough to give proof of presented idea, however, there are some minor changes which should be made in order to improve the clarity and quality of presented work:

We thank the reviewer for the positive evaluation of the manuscript. We have taken into account all your comments and suggestions, since we believe they significantly improve the quality of the manuscript.

Section 2.2: Some dimensions (Fillet radius) of dog-bone specimen are missing.

We agree with your suggestion, thus we have added the radius of the fillet to the dog-bone specimen shown in Figure 1.

Section 2.3-line(141): “rotability” is wrong word if not then what is the meaning of this word. Section 2.3-line(143): Explain “rotatable” word for better understanding of readers

Thank you for your critical assessment of the language used in the manuscript. The word rotability is in fact not correct. The correct word is rotatability as defined by Norman Draper in the encyclopedia of statistics in Quality and Reliability https://doi.org/10.1002/9780470061572.eqr034.  We have added an explanation of what a rotatable design is as follows:

The desired characteristic of each DoE is the independent estimation of the main factors and their interactions, which is a function of the rotatability of the design. The rotatability of the design is accomplished by adding trials so that all design trials (parameters values) are equally distant from the centre of the design (mean values of parameters). In other words, a design is rotatable if it predicts the values of the parameters with the same precision at all points equidistant from the coded origin of the design [56]. A design is rotatable if [55]…

The figure name is not centered with figures (Fig. 1,2-4). The quality of graphs needs improvement (Fig. 7,19).  

Thank you for the comments. Figures 1, 2-4 are centered. The quality of figures 1, 7, 9-21 and 23 has been improved. Also on figures 7 and 23 step of scale for X axis was changed from 4 to 5. On figures 7, 9-21 and 23 the units N/mm2 were changed to N/mm2. On figures 9, 12, 13, 15 and 19, the units oC were changed to °C.

Section 3.1 is well written, however some sections is lacking the flow of sentences like Section 3.4. revise and modify for the improvement of paper quality. 

Thank you very much for the comments, they greatly improve the clarity of the manuscript and as such we have modified Section 2.4 and Section 3.4. The new sections now read as follows:

Section 2.4

Tensile strength (N/mm2), maximal tensile force (N) and tensile modulus (N/mm2) were used as characterisation parameters for the mechanical tensile properties of specimens. Additionally, the mass of the test specimens was measured in order to relate it to the achieved mechanical properties of the tested specimens. Specimens were tested with a deformation speed of 2 mm/min. The results of tensile testing of 3D printed specimens are presented in Table 3.

Section 3.4

In order to set the FFF parameters for the production of specimens with optimal properties, the criteria for optimisation have to be determined (Table 7). Here complex optimisation based on multiple simultaneous criteria was used. The limits of the expected tensile properties, shown in Table 7, are set according to the results obtained from the previous DoE analysed in section 3.3.

The mathematical method for finding the optimum of the observed response surface was the desirability function. The desirability function ranges from 0 to 1, where 1 is the goal of the optimisation. The numerical results of the complex optimisation with the highest desirability function are shown in Table 8.

All three observed FFF parameters (extrusion temperature, flow rate multiplier and layer thickness) have to be maximised within the previously determined processing window in order to maximize the tensile properties of the green specimens. From Figures 19 to 21 it is clear that increasing flow rate multiplier results in the highest increase of all observed tensile properties, while extrusion temperature has the lowest effect on the observed properties. As it was already mentioned, the extrusion temperature only has a strong effect on the polymeric components (i.e 45 % vol of the feedstock); therefore its effects on the mechanical properties are generally at a lower level compared to the properties of pure polymeric FFF specimens [49].
